# Submergence deactivates wound-induced plant defence against herbivores

Hyo-Jun Lee [1,2 ✉], Ji-Sun Park[1], Seung Yong Shin[1], Sang-Gyu Kim [3], Gisuk Lee[3], Hyun-Soon Kim[1,4], Jae Heung Jeon[1] & Hye Sun Cho [1,4]

Flooding is a common and critical disaster in agriculture, because it causes defects in plant growth and even crop loss. An increase in herbivore populations is often observed after floods, which leads to additional damage to the plants. Although molecular mechanisms underlying the plant responses to flooding have been identified, how plant defence systems are affected by flooding remains poorly understood. Herein, we show that submergence deactivates wound-induced defence against herbivore attack in *Arabidopsis thaliana*. Submergence rapidly suppressed the wound-induced expression of jasmonic acid (JA) biosynthesis genes, resulting in reduced JA accumulation. While plants exposed to hypoxia in argon gas exhibited similar reduced wound responses, the inhibitory effects were initiated after short-term submergence without signs for lack of oxygen. Instead, expression of ethylene-responsive genes was increased after short-term submergence. Blocking ethylene signalling by *ein2-1* mutation partially restored suppressed expression of several wound-responsive genes by submergence. In addition, submergence rapidly removed active markers of histone modifications at a gene locus involved in JA biosynthesis. Our findings suggest that submergence inactivates defence systems of plants, which would explain the proliferation of herbivores after flooding.

[1] Plant Systems Engineering Research Center, Korea Research Institute of Bioscience and Biotechnology, Daejeon 34141, Korea. [2] Department of Functional Genomics, KRIBB School of Bioscience, University of Science and Technology, Daejeon 34113, Korea. [3] Department of Biological Sciences, Korea Advanced Institute of Science and Technology, Daejeon 34141, Korea. [4] Department of Biosystems and Bioengineering, KRIBB School of Biotechnology, University of Science and Technology, Daejeon 34113, Korea. ✉email: hyojunlee@kribb.re.kr

Flooding is a widespread agricultural disaster, which causes significant damage and loss to crops. Submergence is a representative event of floods in which water covers aerial plant tissues, leading to oxygen deprivation[1,2]. As oxygen acts as a terminal electron acceptor in the oxidative phosphorylation process during mitochondrial respiration, hypoxia causes energy deficit, which is the major reason for reduced growth and death in plants exposed to hypoxic conditions[2,3]. In addition to hypoxia-induced plant damage, outbreak of herbivores is another phenomenon observed after flooding. Both continuous and periodic flooding trigger an increase in herbivore populations in rice[4]. Elevated ovipositional and feeding activities of insect herbivores are also observed in flooded plants including citrus[5,6]. However, the effects of flooding on plant defence responses to herbivore attack remain largely unknown.

Plant defence against herbivores involves the sensing of mechanical wounding, which is a hallmark of herbivore attack. Early signals, including reactive oxygen species (ROS), $Ca^{2+}$, and glutamate, are produced in the wounded area[7,8]. Glutamate triggers an increase in intracellular $Ca^{2+}$ levels through GLUTAMATE RECEPTOR-LIKE proteins, which are cation-permeable ion channels[8]. The increase in cytosolic $Ca^{2+}$ is propagated to nearby tissues, which activates jasmonic acid (JA) biosynthesis[8,9]. The expression of genes involved in JA biosynthesis, including those encoding LIPOXYGENASE (LOX), ALLENE OXIDE SYNTHASE (AOS), ALLENE OXIDE CYCLASE (AOC), and OXOPHYTODIENOATE-REDUCTASE (OPR), is induced within 1 h of wounding[10–12]. The accumulated JA-Ile directly binds to the F-box protein CORONATINE INSENSITIVE 1 (COI1), which forms a functional $SCF^{COI1}$-JAZ co-receptor complex to trigger the degradation of JASMONATE ZIM-DOMAIN (JAZ) proteins[13]. The activation of the downstream JA signalling by JAZ degradation induces defence responses, including the accumulation of defensive proteins and toxic secondary metabolites[14,15].

Studies on the plant adaptive mechanisms in response to submergence have revealed that plants have developed two distinctive growth strategies to survive submergence. To escape from the water, plants focus their energy into upward growth of stems, which is a beneficial strategy under shallow and short-term submergence conditions[16,17]. However, under long-term submergence, plants tend to reduce growth to save energy and carbohydrate reserves for tolerance[16]. Ethylene plays pivotal roles in both strategies under submergence conditions. Submergence rapidly increases ethylene concentrations in plant cells, because water restricts diffusion of the ethylene gas[2]. Accumulated ethylene induces shoot elongation, adventitious root formation, soluble carbohydrate breakdown, and ethanolic fermentation[18–21]. ETHYLENE INSENSITIVE 2 (EIN2) is a key factor in ethylene signalling by which it protects proteasomal degradation of transcription factors to transmit ethylene signals[22]. EIN2-deficient mutants exhibit reduced expression of hypoxia-inducible genes and decreased survival after submergence[23–25], showing its critical role on hypoxia tolerance.

Hypoxia affects a broad range of gene expressions by changing histone modifications to regulate transcriptional activity. In human cultured cells, hypoxia alters histone-3 lysine-4 trimethylation and H3K36me3, possibly by inhibiting Jumanji-C histone demethylase[26]. A study on the histone modifications in rice showed that submergence changes histone methylation and acetylation at *ADH1* and *PDC1* loci, which are responsive to submergence[27]. Notably, while H3K4 methylation is altered upon short-term submergence, H3 acetylation is increased after long-term submergence. These results indicate that plants recognize duration of submergence and modify transcriptional activity of key genes for survival under submergence conditions.

In this work, we found that submergence rapidly inactivates wound-induced JA biosynthesis in *Arabidopsis*. Plants that had undergone submergence exhibited reduced wound responses and increased susceptibility to herbivores. Rapid ethylene responses upon submergence were shown to be partially responsible for decreased wound responses. Submergence also triggered changes of histone methylation and acetylation at the *OPR3* locus. Our results indicate that submergence inactivates plant wound responses, which possibly explains outbreaks of herbivores after flooding.

## Results

**Submergence inhibits wound-induced gene expressions.** Although the early signals of the wound responses and downstream JA signalling mechanisms have been identified, wound-sensing mechanisms are not well understood. Plant leaves are surrounded by the epidermal cells; thus, internal aeration is tightly controlled by the stomatal apertures[28]. As wounding disrupts the enclosed structures in plant tissues, we hypothesized that an abrupt increase in aeration in plant tissues might be an initial cue for sensing mechanical wounding. To limit aeration in wounded tissues, we submerged the entire seedlings and wounded the cotyledons and leaves immediately in the water. We chose *JAZ*s and *OPR3* genes as representative wound-responsive JA-responsive genes and a JA biosynthesis gene, respectively[29,30]. The expression of *JAZ* and *OPR3* genes was analysed to monitor plant wound responses. Whereas the expression levels of *JAZ7*, *JAZ10*, and *OPR3* were significantly elevated after wounding in the air (*JAZ7*, $P = 0.00005$, Hedges′ $g = 5.80$; *JAZ10*, $P = 0.000005$, Hedges′ $g = 7.63$; *OPR3*, $P < 1.0 \times 10^{-7}$, Hedges′ $g = 14.4$), the induction was largely suppressed when wounding was done in water (Fig. 1a). We hypothesized that, if plants cannot sense wounds in the water, re-aeration would lead to recovery of the wound responses. However, expression of the wound-responsive genes was still suppressed following re-aeration (Fig. 1b). In addition, when seedlings were wounded in the air after submergence, wound responses were not recovered (Fig. 1c). Wound-induced gene expression was also significantly suppressed after submergence in adult plants grown on soil for 3 weeks (Supplementary Fig. 1; *JAZ7*, $P = 0.0073$, Hedges′ $g = 2.68$; *JAZ10*, $P = 0.0067$, Hedges′ $g = 2.72$; *OPR3*, $P = 0.013$, Hedges′ $g = 2.41$). We examined the promoter activity of *JAZ10* gene using *pJAZ10:GUS* seedlings; the results showed that wound-induced *GUS* expression was largely down-regulated after submergence (Fig. 1d). Time-course expression analysis after wounding showed that expression of the *JAZ7*, *JAZ10*, and *OPR3* genes was largely decreased particularly at 1 h after wounding in the seedlings that had undergone submergence (Fig. 1e). Increasing submergence duration of more than 1 h resulted in slight additional suppression in the *OPR3* expression (Fig. 1f). Together, these data suggest that while abrupt aeration would not be the initial cue for wound sensing, experience of submergence negatively affects plant wound responses.

**Submergence suppresses JA biosynthesis to reduce defence against herbivores.** The analysed wound-responsive genes were related to JA biosynthesis and JA signalling[12,31]. To determine whether the submergence affects JA biosynthesis or the JA signalling, we treated plants with methyl JA (MeJA) after submergence. The expression of *JAZ7*, *JAZ10*, and *OPR3* genes was largely induced by MeJA treatment, as reported previously[32] (Fig. 2a). Notably, the expression of *JAZ7* was not altered and that of *JAZ10* was reduced only slightly after submergence. In contrast, MeJA-induced *OPR3* expression was largely suppressed after submergence, similar to the results observed upon

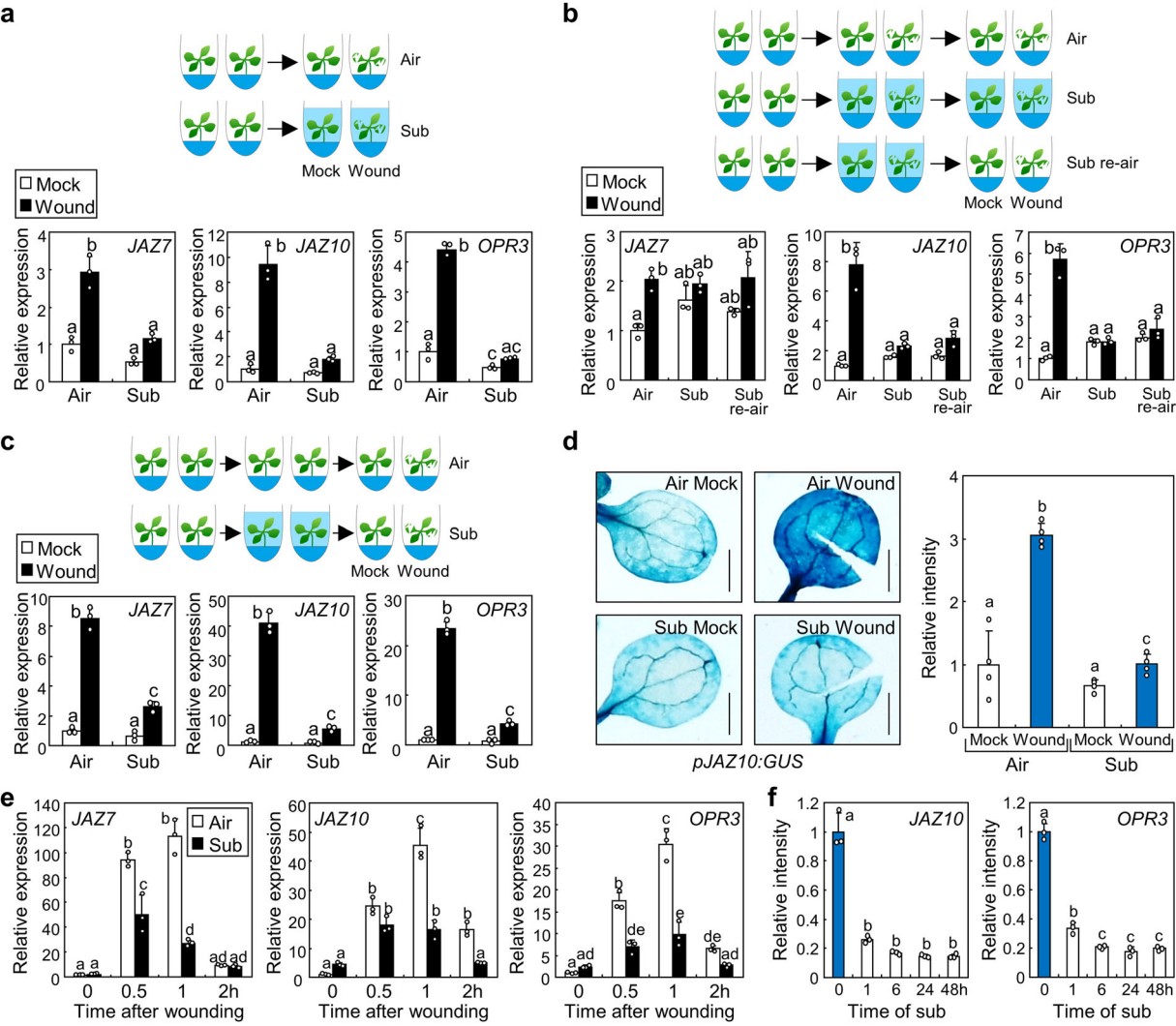

**Fig. 1 Submergence inhibits wound responses.** Ten-day-old seedlings were used. Error bars indicate ±SD. Letters indicate groups that are statistically significantly different from each other ($P < 0.05$, Tukey's test). **a** Effects of submergence on wound responses. Cotyledons and leaves of the Col-0 seedlings were left in the air (air) or transferred into the water (sub), wounded immediately and then harvested after 1 h incubation. Three biological replicates were averaged. **b** Re-aeration does not recover wound responses. The Col-0 seedlings were wounded in the water immediately after submergence and incubated for 1 h. Submerged seedlings were then re-aerated for 1 h before they were harvested (Sub re-air) or were left in water for another hour before harvest (Sub). Three biological replicates were averaged. **c** Submergence inhibits wound responses of re-aerated seedlings. The Col-0 seedlings were submerged for 1 h without wounding and then transferred to air. Re-aerated seedlings were immediately wounded and then harvested after 1 h. Three biological replicates were averaged. **d** Promoter activity of the *JAZ10* gene. Ten-day-old *pJAZ10:GUS* transgenic seedlings were submerged and wounded as described in **c**. Seedlings were fixed with acetone 1 h after the wounding. Intensities of GUS signals were measured using ImageJ software. Four biological replicates were averaged. Scale bars indicate 0.1 cm. **e** Time-course expression of genes after wounding. Seedlings were submerged for 1 h and then transferred to air. Re-aerated seedlings were wounded and then harvested at the indicated time points. Three biological replicates were averaged. **f** Effects of submergence time on wound responses. Seedlings were submerged for the indicated time points and then transferred to air. Re-aerated seedlings were wounded immediately and then harvested after 1 h. Three biological replicates were averaged.

wounding. These data suggest that submergence affects JA biosynthesis. We next analysed the expression of other genes involved in the JA biosynthesis pathways (Fig. 2b). The wound-induced expression of genes encoding AOCs, AOS, and LOXs, which play key roles in JA biosynthesis[33], was significantly reduced after submergence except for *LOX6* (Fig. 2c and Supplementary Fig. 2; *AOC2*, $P < 1.0 \times 10^{-7}$, Hedges' $g = 20.1$; *LOX2*, $P < 1.0 \times 10^{-7}$, Hedges' $g = 15.4$; *AOC1*, $P = 0.000074$, Hedges' $g = 5.31$; *AOC3*, $P = 0.00056$, Hedges' $g = 4.00$; *AOC4*, $P = 0.0055$, Hedges' $g = 4.25$; *AOS*, $P = 0.0000096$, Hedges' $g = 7.00$; *LOX3*, $P = 0.0000053$, Hedges' $g = 7.55$; *LOX4*, $P = 0.0026$, Hedges' $g = 3.17$). Consistently, JA accumulation upon wounding was also decreased after submergence (Fig. 2d), indicating that

submergence affects the transcription of JA biosynthesis genes to inhibit wound-induced JA accumulation.

It is well known that JA mediates plant defence against herbivore attacks[34]. As our data showed that submergence reduces wound-induced JA biosynthesis, we examined plant defence responses after submergence using Brassicaceae specialist, *Pieris rapae* caterpillars. Soil-grown plants were submerged and then transferred to the air before herbivore feeding. We put ~20 caterpillars on 10 plants, but only collected 8~9 caterpillars after 5 days, possibly because several of them moved away from the plants. Larvae fed on the plants that had undergone submergence showed a significantly higher weight than those fed on control plants (Fig. 2e; $P < 1.0 \times 10^{-7}$; Hedges'

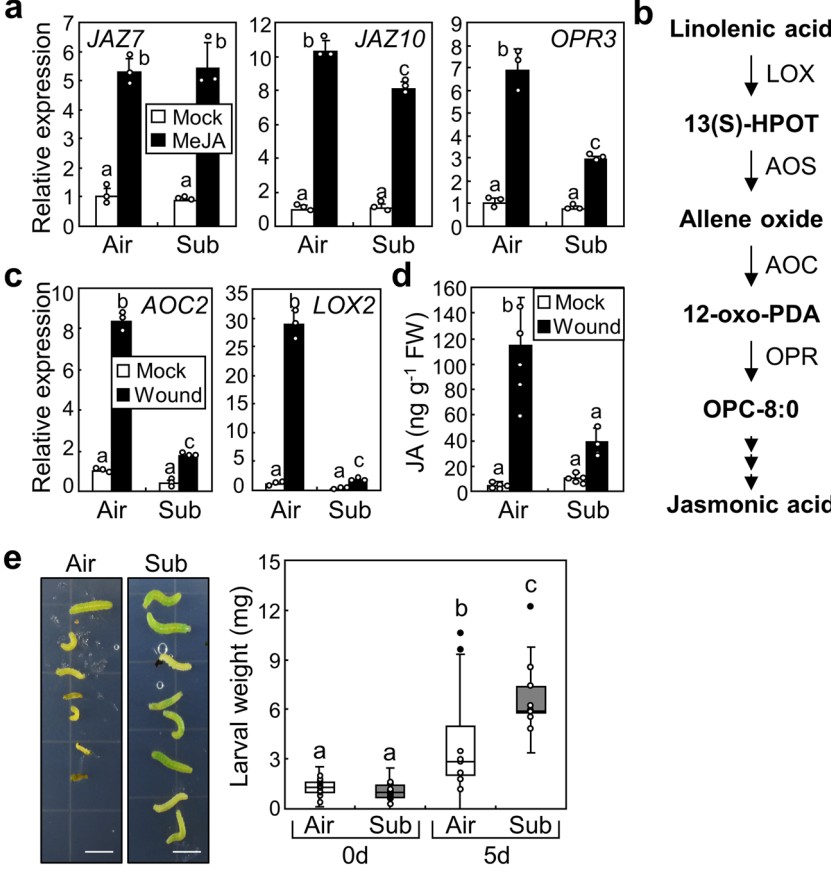

**Fig. 2 Submergence inhibits JA biosynthesis.** Ten-day-old seedlings were used. Error bars indicate ±SD. Letters indicate groups that are statistically significantly different from each other ($P < 0.05$, Tukey's test). **a** Expression of JA-responsive genes after MeJA treatment. The Col-0 seedlings were submerged for 1 h and then transferred to the air. Re-aerated seedlings were treated with 25 µM MeJA. Whole seedlings were harvested 1 h after the treatment. Three biological replicates were averaged. **b** JA biosynthesis pathways. 13(S)-HPOT, (9Z,11E,15Z,13S)-13-hydroperoxy-9,11,15-octadecatrienoic acid; 12-oxo-PDA, 12-oxo-10,15(Z)-octadecatrienoic acid; OPC-8:0, 3-oxo-2(2'(Z)-pentenyl)-cyclopentane-1-octanoic acid. **c** Expression of JA biosynthesis genes after the submergence. The Col-0 seedlings were submerged and wounded as described in Fig. 1c. Three biological replicates were averaged. **d** JA accumulation after the wounding. The Col-0 seedlings were submerged and wounded as described in Fig. 1c. Five biological replicates were averaged. **e** Herbivore-feeding assay. Four-week-old Col-0 plants were submerged for 1 h and then transferred to the air. Two caterpillars (larval stage L1) of the Brassicaceae specialist *P. rapae* were released onto a single reaerated 4-week-old plant. Larval weights were measured before and 5 days after the caterpillar feeding ($n = 8$-24). The box plot shows median and quartiles. Scale bars indicate 0.5 cm.

$g = 3.72$), indicating that defence against herbivores is suppressed after submergence.

**Hypoxia inhibits wound responses after re-aeration**. Wounding triggers the accumulation of early signalling molecules in the wounded leaves; thereafter, the signals are propagated to other tissues, leading to JA accumulation[7–9]. Therefore, we analysed whether submergence affects early wound responses. Hydrogen peroxide ($H_2O_2$) is one of the initial signals of wounding[7], but submergence did not alter $H_2O_2$ accumulation after wounding (Supplementary Fig. 3a). Increase in the concentration of cytosolic $Ca^{2+}$ is another important event in the early wound responses[9]. We analysed the expression of *TCH2* and *TCH3*, which are molecular markers of cytosolic $Ca^{2+}$ increase[35]. The expression of these genes was induced upon wounding, but submergence without wounding also elevated the expression of *TCH2* and *TCH3* (Supplementary Fig. 3b). Wounding after submergence did not lead to additional increase in gene expressions. As the wound-induced cytosolic accumulation of $Ca^{2+}$ is controlled by glutamate[8], we measured glutamate levels using *35S:CHIB-iGluSnFR* glutamate reporter plants, which have been previously described[8]. As with molecular $Ca^{2+}$ markers, there

was an accumulation of glutamate after submergence (Supplementary Fig. 3c). Again, wounding did not further elevate glutamate levels after submergence. If saturation of glutamate accumulation and the corresponding $Ca^{2+}$ increase diminished wound responses in submerged plants, then high concentrations of glutamate treatment should have similar results. However, the wound-induced expression of *JAZ10* and *OPR3* genes was still observed in both the mock- and glutamate-treated plants (Supplementary Fig. 3d). Submergence-mediated decrease in the expression of wound-responsive genes was also observed after glutamate treatment. Together, these data indicate that suppression of wound responses after submergence is not directly related with early wound signals.

As the solubility of oxygen is lower in water than in air, hypoxia responses are induced in submerged plants[36]. To investigate whether the reduction of wound responses after submergence was related to oxygen deprivation, we built a system for incubation of plants under a continuous flow of argon gas (Fig. 3a). Incubation of the seedlings in a transparent box filled with argon gas for 2 h triggered induction of a gene encoding ALCOHOL DEHYDROGENASE 1 (ADH1), a molecular marker for hypoxia[36] (Fig. 3b). After argon gas treatment, seedlings were transferred to ambient air and then wounded. The expression of

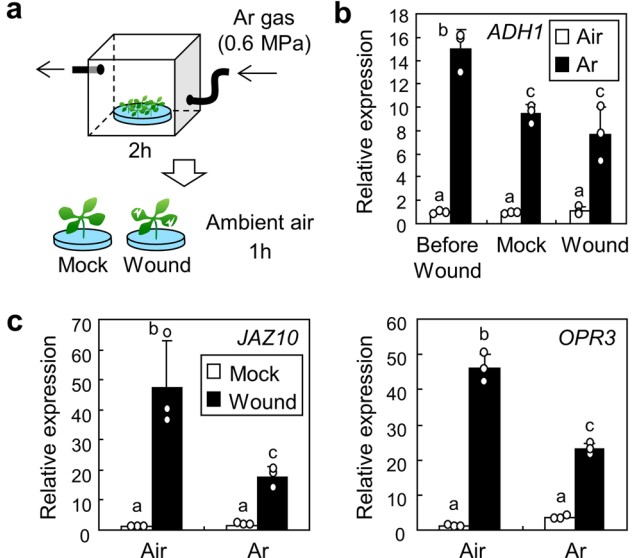

**Fig. 3 Hypoxia suppresses wound responses.** Ten-day-old seedlings were used. Error bars indicate ±SD. **a** Schematic illustration of the experiment. The Col-0 seedlings were incubated in the transparent glove box filled with Ar gas for 2 h. Seedlings were then transferred to the ambient air and wounded. **b** Expression of the *ADH1* gene. Whole seedlings treated as described in **a** were harvested before and 1 h after the wounding. Expression of the *ADH1* gene was analysed as a hypoxia marker. Three biological replicates were averaged. Letters indicate groups that are statistically significantly different from each other ($P < 0.05$, Tukey's test). **c** Expression of the wound-responsive genes. Whole seedlings treated as described in **a** were harvested 1 h after the wounding. The *JAZ10* and *OPR3* genes were analysed. Letters indicate groups that are statistically significantly different from each other ($P < 0.05$, Tukey's test).

*JAZ10* and *OPR3* genes was analysed 1 h after wounding. Expression of wound-responsive genes was significantly reduced after hypoxia (Fig. 3c; *JAZ10*, $P = 0.0082$, Hedges' $g = 2.62$; *OPR3*, $P = 0.0000048$, Hedges' $g = 7.69$), suggesting that oxygen availability is related with suppression of plant wound responses.

**Ethylene is involved in submergence-mediated suppression of wound responses.** For detailed examination of the effects of submergence on plant wound responses, we submerged seedlings for different time durations before wounding. Submergence for even 1 min resulted in inhibition of wound responses and more than 10 min of submergence strongly suppressed the expression of wound-responsive genes (Fig. 4a). Less than 10 min of submergence is not sufficient to induce hypoxia in plants under light conditions[37,38]. In this work, plants were submerged in the light. In agreement with previous studies, expression of a hypoxia marker gene, *ADH1*, was not induced during short-term submergence (Fig. 4a). We thus speculated that hypoxia is not the main reason for suppression of wound responses by short-term submergence. To find clues, we analysed additional JA-responsive genes in the plants that had undergone submergence. Although expression of *JR1* and *VSP1* genes exhibited similar pattern to that of *JAZ10*, expression of *PR* genes was not largely altered by both wound and submergence in our work (Supplementary Fig. 4). Notably, expression of ethylene-responsive factors (*ABR1* and *ERF4*) and ethylene- and JA-responsive *PDF1.2* was significantly induced by 1 h submergence[11,39,40] (Fig. 4b and Supplementary Fig. 4; *ABR1*, $P = 0.0000022$, Hedges' $g = 13.2$; *ERF4*, $P = 0.00039$, Hedges' $g = 5.18$; *PDF1.2*, $P = 0.033$, Hedges' $g = 3.67$). Ethylene is a key phytohormone that mediates plant submergence responses[2]. To examine cross-talks between ethylene

signals and wound responses, we investigated wound responses after submergence using ethylene-insensitive *ein2-1* mutant. Wound-induced expression of *AOC2*, *JAZ7*, and *VSP1* genes after submergence was partially restored in *ein2-1* mutant, but other JA-biosynthesis and JA-responsive genes were not significantly altered by *ein2-1* mutation (Fig. 4c). These data suggest that ethylene partially mediates early submergence responses to inhibit plant wound responses, but there are other unknown regulators in these processes.

**Submergence removes active histone modifications at the *OPR3* locus.** Hypoxia triggers changes of histone modifications in both animals and plants[26,27]. We therefore analysed histone modifications at the *OPR3* locus using chromatin immunoprecipitation (ChIP) assays (Fig. 5a). H3K4me2 and acetylation of histone H3 (H3Ac) are considered as markers of active transcription[41,42]. ChIP assays showed that H3K4me2 was significantly decreased by submergence in both mock- and wound-treated seedlings (Fig. 5b; P6-mock, $P < 1.0 \times 10^{-7}$, Hedges' $g = 13.7$; P6-wound, $P = 0.0000001$, Hedges' $g = 15.4$). However, changes of H3K4me2 in mock-treated seedlings would be a technical error, because H3K4me2 was also reduced by submergence at the control *ACTIN 7* (*ACT7*) locus (Fig. 5b). Otherwise, H3Ac was significantly suppressed by submergence in both mock- and wound-treated seedlings (P4-mock, $P = 0.0000001$, Hedges' $g = 10.8$; P4-wound, $P < 1.0 \times 10^{-7}$, Hedges' $g = 11.6$), while that did not exhibit any change in the control regions (Fig. 5c). We also analysed trimethylation of histone H3 lysine 9 (H3K9me3), which is associated with heterochromatin[43], but the results did not yield meaningful data (Supplementary Fig. 5). These results suggest that submergence epigenetically regulates the *OPR3* gene to inactivate gene expression before wounding. To find genes responsible for submergence-mediated histone modifications, we analysed expression of the *OPR3* gene in the mutants that are deficient in a histone deacetylase (*axe1-5*)[44], a histone demethylase (*jmj30-2*)[45], and histone methyltransferases (*atx1-2 atx2-1*)[46]. However, expression of the *OPR3* gene in these mutants was similar to that in Col-0 seedlings (Supplementary Fig. 6). We next treated Col-0 seedlings with trichostatin A (TSA), which is a histone deacetylase inhibitor, to examine the role of histone deacetylases in these responses. The expression of the *OPR3* and *JAZ10* genes in the seedlings that had undergone submergence was not altered by TSA treatment until 1 h after wounding (Fig. 5d). However, wounding slightly but significantly elevated expression of *OPR3* and *JAZ10* genes at 2 h after wounding in TSA-treated seedlings (*OPR3*, $P = 0.0019$, Hedges' $g = 24.1$; *JAZ10*, $P = 0.0000002$, Hedges' $g = 21.5$), whereas changes upon wounding were not significant in mock-treated seedlings (Fig. 5d). These results suggest that the histone deacetylase affects submergence-mediated inhibition at late stage of wound responses. All together, our findings suggest that even a short-term experience of submergence triggers repression of plant defence responses against herbivores. Active histone modifications at the *OPR3* locus would enable wound-induced gene expression in plants grown under ambient air (Fig. 5e). However, submergence induces ethylene and other unknown responses to inhibit JA biosynthesis. In addition, submergence removes active histone modifications; thus, subsequent wounding cannot efficiently induce the expression of JA biosynthesis genes, resulting in reduced defence against herbivores (Fig. 5e).

**Discussion**

Many studies have identified the molecular signalling pathways for a single abiotic or biotic stress, but how previous environmental stresses affect upcoming biotic challenge is largely

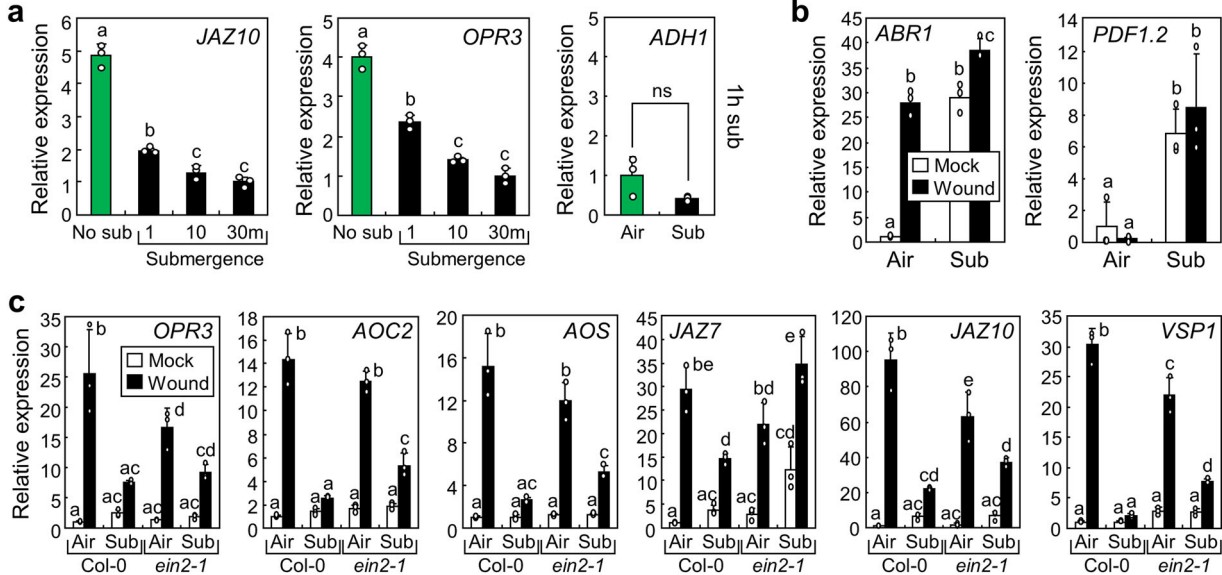

**Fig. 4 Ethylene partially mediates submergence responses.** Ten-day-old seedlings were used. Error bars indicate ±SD. **a** Effects of submergence time on wound responses. The Col-0 seedlings were submerged for the indicated time periods and then transferred to the air. Re-aerated seedlings were wounded immediately and harvested 1 h after the treatment. For *ADH1*, the Col-0 seedlings were harvested right after 1 h submergence. Three biological replicates were averaged. Letters indicate groups that are statistically significantly different from each other ($P < 0.05$, Tukey's test). NS, not significant. **b** Expression of wound- and ethylene-responsive genes. The Col-0 seedlings were submerged and wounded as described in Fig. 1c. Three biological replicates were averaged. Letters indicate groups that are statistically significantly different from each other ($P < 0.05$, Tukey's test). **c** Expression of wound-responsive genes in *ein2-1* mutant. The Col-0 and *ein2-1* seedlings were submerged and wounded as described in Fig. 1c. Three biological replicates were averaged. Letters indicate groups that are statistically significantly different from each other ($P < 0.05$, Tukey's test).

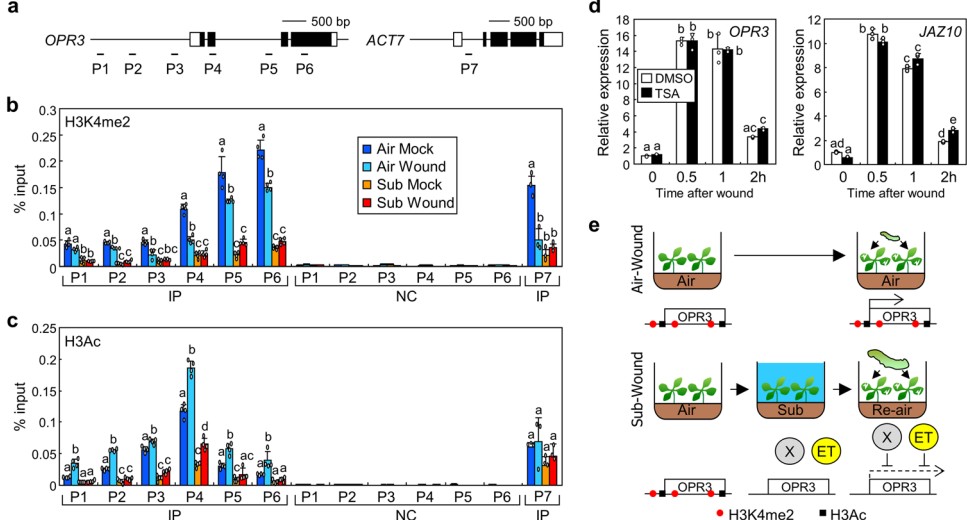

**Fig. 5 Submergence affects histone modifications. a** Genomic structure of the *OPR3* gene. Black boxes, exons; white boxes, untranslated regions. Sequence elements used for ChIP assays are annotated as P1 to P7. **b, c** Histone modifications at *OPR3* locus. Ten-day-old Col-0 seedlings were submerged for 1 h and then transferred to the air. Re-aerated seedlings were wounded immediately and harvested 1 h after the treatment. Anti-H3K4me2 (**b**) and anti-H3Ac (**c**) antibodies were used for immunoprecipitation. Three to four biological replicates were averaged. Error bars indicate ±SD. Letters indicate groups that are statistically significantly different from each other ($P < 0.05$, Tukey's test). It is noteworthy that the data were statistically analysed separately for individual sequence elements by one-way ANOVA. Individual data points were marked in IP. IP, immunoprecipitation; NC, negative control. **d** Effects of TSA on wound-induced gene expressions after submergence. The Col-0 seedlings grown on DMSO- or TSA-containing MS-agar media were submerged for 1 h and then transferred to the air. Re-aerated seedlings were wounded and harvested at the indicated time points after the treatment. Three biological replicates were averaged. Letters indicate groups that are statistically significantly different from each other ($P < 0.01$, Tukey's test). Error bars indicate ±SD. It is noteworthy that DMSO is used for mock treatment. **e** Schematic model for wound-induced defence responses after the submergence. In the ambient air, wounding induces expression of JA biosynthesis genes for defence against herbivores. When plants undergo submergence, ethylene signalling was activated and histone modifications were changed from the active to inactive states. There would be other unknown regulators (X) activated by submergence to inhibit wound responses. These events would block wound-induced gene expressions, resulting in reduced herbivore resistance.

unknown. This work found that experience of submergence causes negative effects on plant resistance to herbivores by suppressing JA biosynthesis. Why do plants inactivate defence systems against herbivores after submergence? Previous studies have reported that JA antagonizes salicylic acid (SA) signalling. Activation of JA signalling results in inhibition of SA-dependent gene expressions in tobacco[47]. In addition, JA-insensitive coi1 mutant exhibits enhanced expression of SA-dependent genes and resistance to Pseudomonas syringae[47]. Thus, it is possible that the JA signalling pathways are inhibited in plants to improve SA-dependent pathogen resistance after submergence. In support of this idea, a recent study has reported that prior submergence enhances resistance to bacterial pathogen P. syringae by upregulating expression of a WRKY22 gene in Arabidopsis[38]. WRKY22 is a SA-responsive gene and high expression of WRKY22 causes negative effects on defence against aphids[48]. Further research on the JA-SA cross-talks would provide important clues regarding the molecular mechanisms on submergence-mediated suppression of plant wound responses.

A recent study has reported that AOS and hydroperoxide lyase (HPL)-mediated metabolic changes confer waterlogging tolerance in Arabidopsis[49]. Waterlogging induces production of 12-oxo-PDA (12-oxo-10,15(Z)-octadecatrienoic acid), which is a precursor of JA. However, JA content is not changed by waterlogging stress[49]. Our study also showed that submergence does not affect JA levels in the absence of wound stress (Fig. 2d). It is likely that plants induce JA-independent metabolic changes for tolerance to waterlogging stress by regulating AOS and HPL activities. On the other hand, plants reduce wound responses by suppressing JA biosynthesis possibly for increasing JA-antagonistic responses including pathogen resistance after submergence.

Ethylene is a key phytohormone in plant submergence responses. Diffusion rate of ethylene in water is lower than that in ambient air, thus submergence induces rapid ethylene responses in plants[2,36]. Our data suggested that the ethylene response is involved in submergence-mediated suppression of wound responses. Wound-induced expressions of JA-biosynthesis genes and JA-responsive genes were significantly suppressed by submergence in Col-0 seedlings, but those of several genes including AOC2, JAZ7, and VSP1 were partially restored in the ethylene-insensitive ein2-1 mutant. EIN2 is a key molecule in ethylene signalling. In the presence of ethylene, the ethylene receptors deactivate a negative regulator CONSTITUTIVE TRIPLE RESPONSE 1 to relieve suppression of EIN2[22]. EIN2 protects downstream transcription factors including EIN3 and EIN3-LIKE 1 (EIL1) from the proteasomal degradation mediated by EIN3-BINDING F BOX PROTEIN 1 (EBF1) and EBF2[22,50]. Therefore, it is possible that EIN2 mediates submergence-induced ethylene signalling to block JA biosynthesis. In support of this idea, previous studies have reported the JA-ethylene cross-talks at the molecular level. EIN3/EIL1 transcription factors physically interact with MYC2, a key transcription factor in JA signalling pathways[51]. The EIN3/EIL1-MYC2 interaction suppresses transcriptional activities of both EIN3/EIL1 and MYC2, showing antagonistic cross-talks between JA and ethylene. Because MYC2 activates JA-responsive promoters of JA biosynthesis genes[52], the ethylene-mediated suppression of MYC2 could be the reason for the reduced expression of JA biosynthesis genes after submergence.

Our data showed that submergence rapidly suppresses plant wound responses with no signs of oxygen deprivation. These results indicate a crosstalk between wound responses and early signals of submergence rather than oxygen deprivation. Mutation of EIN2 partially restored wound responses after submergence, but still the expression of wound-responsive genes was largely reduced (Fig. 4c). Previous studies proposed that EIN2-independent pathways would be involved in ethylene signalling. Ethylene-insensitive phenotype of the ein2 mutant is restored by reducing JA levels[53]. Also, EIN2 TARGETING PROTEIN 1 (ETP1) and ETP2-deficient mutants exhibit constitutive stabilization of EIN2 but are still responsive to ethylene[54], raising the possibility that an EIN2-independent ethylene pathway could be related with submergence-mediated suppression of wound responses. Otherwise, submergence could activate ethylene-independent regulators to inhibit wound responses. It is possible that the synergistic action of EIN2 and other unidentified regulators is required to fully suppress wound responses upon submergence.

Submergence-mediated removal of active histone modification at the OPR3 locus led us to investigate upstream regulators for the epigenetic regulation. However, the key genes for the histone modifications under submergence conditions were not identified. TSA treatment assays showed that histone deacetylases are at least partially involved in suppression of late stage wound responses after submergence (Fig. 5d), but the effects of TSA were not enough to fully explain molecular mechanisms of a crosstalk between submergence and wound responses. These results suggest that multiple histone modifications might be redundantly involved in these responses. Genetic analyses using mutants deficient in multiple histone modifications would be required to find clues for the detailed mechanisms.

## Methods

**Plant materials and growth conditions**. Arabidopsis thaliana ecotype Columbia (Col-0) was used for the assays. The atx1-2 atx2-1 (N71692) seeds were obtained from the Nottingham Arabidopsis Stock Centre (NASC, Nottingham, UK). The ein2-1 and axe1-5 seeds were a gift from Dr. Young-Joon Park and previously described[44,55]. The jmj30-2 seeds were a gift from Dr. Pil Joon Seo and previously described[45]. Seeds were surface-disinfected using 70% ethanol and then incubated at 4 °C for 3 days. After stratification, seeds were transferred to a growth room set at 24 °C and at 50% humidity. Seedlings were grown on half-strength Murashige and Skoog-agar (MS-agar) plates containing 0.7% agar without sucrose under long days (16 h light and 8 h dark cycles). Plates were placed horizontally in the growth room. White light with an intensity of about 100 $\mu$mol m$^{-2}$ s$^{-1}$ was applied using fluorescent FL40EX-D tubes (Focus, Bucheon, Korea).

**Submergence treatment**. Col-0 seedlings grown on MS-agar plates for 10 days under long days were used. Lids of the plates were opened before submergence. Seedlings were submerged in distilled water for the indicated time points in the figure legends. During submergence, plates were placed horizontally in the water-filled transparent box without lids. Submergence treatment was performed in the growth room in light. After submergence, plates were transferred to air and waters in the plates were poured out and lids were closed. To analyse wound responses, cotyledons and first leaves of the seedlings were wounded with scissors right after submergence.

**Reverse-transcription quantitative PCR**. Total RNA was extracted from the seedlings after appropriate treatments using RNeasy Plant Mini Kit (QIAGEN) with an on-column DNase I treatment. Reverse-transcription reaction was performed to produce cDNA using TOPscript cDNA Synthesis Kit (Enzynomics). Reverse-transcription quantitative PCR (RT-qPCR) reactions were carried out in 96-well plates with a CFX Connect Real-Time PCR Detection System (Bio-Rad) using TOPreal qPCR PreMIX (Enzynomics). The primers used in RT-qPCR reactions were listed in Supplementary Table 1. A reference gene UBC21 (AT5G25760) was included as an internal control for normalization. The comparative $\Delta\Delta C_T$ method was used to evaluate relative quantities of each amplified product in the samples. The threshold cycle ($C_T$) was automatically determined for each reaction by the system set with default parameters.

**Histochemical assays**. A 2112 bp fragment containing the promoter and the 5′-untranslated region of the JAZ10 gene was fused to the 5′-end of the β-glucuronidase (GUS)-coding gene in the vector pCAMBIA1305.2[29]. The fusion construct was transformed into Col-0 plants. We obtained three independent second generation (T2) pJAZ10:GUS transgenic plants and confirmed wound-responsive GUS expression in all transgenic lines. A single line was used for histochemical GUS assays to observe effects of submergence. Transgenic seedlings grown on MS-agar plates for 10 days under long days were submerged for 1 h and then transferred to air. First leaves of reaerated plants were wounded and incubated at 24 °C under the light for 1 h and then fixed in 90% acetone for 20 min on ice.

Seedlings were washed twice with rinsing solution containing 50 mM sodium phosphate pH 7.2, 0.5 mM $K_3Fe(CN)_6$, and 0.5 mM $K_4Fe(CN)_6$. Fixed seedlings were incubated at 37 °C for 16 h in staining solution containing 2 mM X-Gluc (Duchefa) in the rinsing solution. After the staining, seedlings were incubated in 70% ethanol for 4 h. For 3,3′-diaminobenzidine (DAB) staining, seedlings were submerged for 1 h and then transferred to air. First leaves of reaerated plants were wounded in the air. After 30 min, seedlings were incubated in DAB staining solution containing 1 mg/ml DAB, 10 mM $Na_2HPO_4$, and 0.05% (v/v) Tween 20 at 24 °C for 16 h in darkness. After staining, DAB staining solution was removed and the seedlings were incubated in 70% ethanol for 4 h. Plant materials were mounted on slide glasses and were photographed using Nikon D5600 digital camera (Nikon).

**Hypoxia treatment**. Transparent glove box containing two holes for gas flow (GBI) was used for hypoxia treatment. Col-0 seedlings grown on MS-agar plates at 24 °C for 10 days under long days were placed in the glove box with lid open and the pressure was held at 0.6 MPa with flow of 99.999% (v/v) argon. Seedlings were incubated in the box for 2 h under light conditions and then were wounded in ambient air.

**Pharmacological treatment**. For MeJA treatment, Col-0 seedlings grown on MS-agar plates at 24 °C for 10 days under long days were submerged for 1 h and then sprayed with 25 μM MeJA (Sigma-Aldrich/Merck). For glutamate treatment, Col-0 seedlings grown on MS-agar plates were submerged for 1 h and then sprayed with 100 mM L-glutamate (Sigma-Aldrich/Merck). For TSA treatment, Col-0 seedlings were grown on MS-agar plates for 2 days. Seedlings were then transferred to other MS-agar plates containing 5 μM TSA and further grown for 8 days. As TSA is dissolved in dimethyl sulfoxide (DMSO), DMSO-treated seedlings were used as a mock control. Seedlings were submerged for 1 h and then transferred to air. Re-aerated seedlings were wounded and then harvested at the annotated time points after wounding.

**Confocal microscopy**. The iGluSnFR fragment (from Addgene plasmid #41732) was amplified by PCR and inserted into pBA002 vector containing CaMV 35S promoter and NOS terminator. The 63 bp of chitinase (AT3G12500) signal peptide was fused in-frame to the 5′-end of the iGluSnFR fragment, forming *35S:CHIB-iGluSnFR* construct[8]. The construct was transformed into Col-0 plants. We obtained three independent second generation (T2) *35S:CHIB-iGluSnFR* transgenic plants and confirmed wound-responsive iGluSnFR fluorescence signals in all transgenic lines. A single line was used for detailed analysis to observe effects of submergence. Transgenic seedlings were submerged for 1 h and then wounded in the air. After 5 min, seedlings were placed on slide glasses and were subjected to fluorescence imaging using an LSM 800 confocal microscope (Carl Zeiss). Fluorescence images were analysed using ZEN 2.5 LITE software. Fluorescence intensities were measured using ImageJ software.

**ChIP assays**. Col-0 seedlings grown on MS-agar plates at 24 °C for 10 days under long days were used for ChIP assays. Seedlings were submerged in distilled water for 1 h and then transferred to air. Cotyledons and first leaves of reaerated plants were wounded in air. After 1 h, seedlings were vacuum infiltrated with 1% (v/v) formaldehyde for cross-linking. The cross-linking was quenched by adding glycine. Plant materials were then ground in liquid nitrogen and resuspended in nuclear extraction buffer containing 1.7 M sucrose, 10 mM Tris-Cl pH 7.5, 2 mM $MgCl_2$, 0.15% (v/v) Triton X-100, 5 mM β-mercaptoethanol, 0.1 mM phenylmethylsulfonyl fluoride, and protease inhibitor cocktail tablets (Sigma-Aldrich/Merck). Nuclear fraction was isolated by centrifuge at $16,000 \times g$ for 1 h at 4 °C. The nuclear fraction was lysed with lysis buffer containing 50 mM Tris-Cl pH 8.0, 0.5 M EDTA, 1% (w/v) SDS, and protease inhibitor cocktail tablets. Solutions were then sonicated to fragment chromatins into 0.5 to 0.8 kb. The sonicated solutions were centrifuged at $16,000 \times g$ for 10 min at 4 °C. The supernatants were transferred to new tubes and diluted (1:10) in ChIP dilution buffer containing 1.1% (v/v) Triton X-100, 1.2 mM EDTA, 16.7 mM Tris-Cl pH 7.5, and 167 mM NaCl. The 10% (v/v) of the diluted solutions were used as inputs. Anti-H3K4me2, anti-H3Ac, and anti-H3K9me3 antibodies were added for immunoprecipitation of histone-DNA complexes. Chromatin solutions without antibodies were used as negative controls. The precipitates were collected using protein G magnetic beads (Bio-Rad) and then eluted from the beads to isolate DNA fragments. DNAs in inputs, negative controls, and immunoprecipitated samples were purified using Wizard® SV gel and PCR clean-up system (Promega). Genomic DNA enriched in the chromatin preparations was analysed using qPCR. The qPCR signals derived from the negative controls and immunoprecipitated samples were normalized using those derived from the inputs. The primers used are listed in Supplementary Table 1.

**Herbivore-feeding assays**. Col-0 plants grown in soil at 24 °C for 4 weeks under long days were submerged in distilled water for 1 h before releasing Brassicaceae specialist *P. rapae* caterpillars (Larval stage L1) (Hampyeongnabi). Larval weights were measured before and 5 days after. Soft paintbrush was used to move caterpillars. Twenty caterpillars were released to each experiment and seven to ten survived caterpillars were weighed after 5 days.

**Statistics and reproducibility**. All statistical methods are annotated in the figure legends. Each experiment was repeated at least twice with similar results. The numbers of biological replicates in each assays are also indicated in the figure legends. ImageJ software was used for quantification of GUS, DAB, and iGluSnFR signals. One-way analysis of variance with post hoc Tukey's test was performed using Rstudio software. For gene expression analysis, three biological replicates with ~10 seedlings for each replicate were used. For analysing GUS signals, four biological replicates with a single leaf for each replicate were used. For measuring JA content, five biological replicates with ~100 mg of seedlings for each replicate were used. For herbivore-feeding assay, weights of 8 to 24 caterpillars were analysed. For ChIP assays, four biological replicates with ~20 seedlings for each replicate were used. Note that three replicates were used for analysing P7 locus as a negative control. Bar and box plots were made using Microsoft Excel. Individual data points were included in all plots. In box plots, each box extends from the first quartile to the third quartile values of the data, and centre lines show median values. Whiskers indicate 1.5 times the interquartile range. Black dots indicate outliers.

**Reporting summary**. Further information on experimental design is available in the Nature Research Reporting Summary linked to this paper.

## Data availability

All data are available in the main text or the Supplementary Information. All the source data for graphs in Figures and Supplementary Information are presented in Supplementary Data 1–11. The *Arabidopsis* Genome Initiative locus codes for the genes discussed in this study are as follows: JAZ7 (AT2G34600), JAZ10 (AT5G13220), OPR3 (AT2G06050), AOC2 (AT3G25770), LOX2 (AT3G45140), AOC1 (AT3G25760), AOC3 (AT3G25780), AOC4 (AT1G13280), AOS (AT5G42650), LOX3 (AT1G17420), LOX4 (AT1G72520), LOX6 (AT1G67560), TCH2 (AT5G37770), TCH3 (AT2G41100), ADH1 (AT1G77120), ABR1 (AT5G64750), PDF1.2 (AT5G44420), JR1 (AT3G16470), PR2 (AT3G57260), PR5 (AT1G75040), VSP1 (AT5G24780), ERF4 (AT3G15210), ACT7 (AT5G09810), and UBC21 (AT5G25760).

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

## Acknowledgements

We thank Dr. Young-Joon Park and Dr. Pil Joon Seo for sharing seeds, Dr. Kwangsoo Shin for building hypoxia treatment systems. This work was supported by the Basic Research Program provided by the National Research Foundation of Korea (NRF-2019R1C1C1002045), New Breeding Technologies Development Program (Project number PJ01480201) provided by the Rural Development Administration of Korea, and the KRIBB Research Initiative Program (KGM5372012).

## Author contributions

H.-J.L. conceived and designed the experiments. H.-J.L. performed gene expression analysis, DAB staining, and confocal microscopy. H.-J.L. and J.-S.P. performed ChIP assays. H.-J.L. and S.Y.S. performed herbivore-feeding assay. S.-G.K. and G.L. measured JA content. H.-S.K. and J.H.J. provided experimental tools and equipment. H.-J.L. prepared the manuscript with the contributions of H.-S.K. and H.S.C.

## Competing interests

The authors declare no competing interests.
