## [Peer Review File · Communications Biology]

Reviewers' comments:

Reviewer #1 (Remarks to the Author):

This submission represents a work on Arabidopsis plants under submergence and an influence of this treatment on herbivore response. In principle, this topic is relatively new and interesting for many researchers. However, in its present form this manuscript requires substantial improvements.

In general, this work focuses only on a very small set of genes that are involved in wounding/pathogen response. However, there are many more factors, as well as wound-responsive genes at different stages of the response, which are ignored here. I miss an analysis of wound-responsive genes, despite JA metabolism and action, for example PR proteins or PDFs. Also, the wound response has certain timing, and you only look at a 1-h-timepoint.

This work could make use of existing transcriptome studies on hypoxia and submergence. It remains unclear why you chose those few genes at all, and not others. How is their expression in other hypoxia/ submergence experiments? Are those the only pathogen-related genes differentially expressed? A short look into published data does not reveal those few genes as mostly affected. Indeed, these genes are often not regulated in similar stress studies.

Another major problem is the timing of stress. Obviously, the authors see an effect already after 1 min of submergence, and used a treatment up to 1 h. However, this is not a natural situation, where flooding often occurs for many hours or days. You need to include more time-points (e.g., 24 and 48 h) to get a realistic situation.

The authors claim that low-oxygen/ hypoxia might be involved. However, after 1 min, there is no hypoxia in submerged leaves, even if submergence was done in darkness. Within 1 h of submergence in darkness, oxygen levels decline, but not up to severe hypoxia (see Lee et al. 2011, Vashisht et al. 2011). If submergence is done in illumination, there is not hypoxia at all. The authors do not specify the light levels during submergence and of control plants, and they also do not mention at what time (related to start of daylight) the stress was applied.

In line with this, if you expect hypoxia, you should demonstrate this by analyzing for example ADH1 expression in your submergence treatments, as you did for the hypoxia gassing.

Another plant hormone that is important under submergence and increases much faster than oxygen declines is ethylene. This would be a much more obvious signal for plants, rather than oxygen deficiency. Interaction between ethylene and JA might be present.

The experiments are sometimes not easy to understand. The drawings are helpful, but not always accurate, and the Mat&Meth section is also not complete. For example, lines 179-180 only state wounding after de-submergence, but some data apparently are from wounding under water. How was the timing there? How long were the plants under water before wounding in water? Only a few seconds, or a few minutes? In Fig. 1b, do you also have an air/re-air time point (i.e., 2 h after wounding)? The wounding response quickly declines, and maybe it already went down during this time.

The manuscript ignores previous publications that also deal with pathogens and submergence, and that both show a positive effect of submergence on pathogen tolerance. Hsu et al. 2013 demonstrate enhanced bacterial resistance after flooding, and Baumler et al. 2019 describe submergence-induced expression of ERF#111, a gene that strongly responds to wounding.

Regarding ChIP assays, should you not use another negative control, i.e. a promoter that does not respond to your treatment, rather than relate the quantification to the input fraction?

Line 158, please specify "pathogen infection" (i.e., which pathogens).

Materials and Methods, did your MS medium contain sugars? Were the plates placed vertically or horizontally in the growth chambers? How was the positioning during submergence? How much agar was in the medium? Was submergence and hypoxia treatment in light or darkness?

Your transgenic lines (line 16, line 225), which generation did you use? How many independent lines did you use?

In Figure S3, you did not do ANOVA although these are similar data as in the main manuscript. Please use appropriate statistics throughout the figures. Figure S3d, what does "DW" mean? Did you use a mock spraying treatment?

Reviewer #2 (Remarks to the Author):

The manuscript entitled 'Submergence deactivates wound-induced plant defence against herbivores' by Lee et al reported the crosstalk between submergence/hypoxia and the JA-mediated wound response in Arabidopsis. The authors revealed that upon submergence or hypoxia, some JA biosynthesis or signaling pathway genes could not be properly upregulated, resulting in the inactivation of wound response. Furthermore, the authors showed that histone modifications were changed on JA gene loci, resulting in a "memory" of submergence/hypoxia for inactivation of wound response in a relatively long period. Overall, the manuscript was well organized and presents many interesting data. Although the authors did not show the molecular mechanism of how submergence/hypoxia affects JA gene expression and epigenetic changes, this is still a very interesting discovery and provides new insights into the field. Here I have several suggestions for the authors to improve the manuscript.

Specific points:

1) The authors may add more introductions to the current knowledge on the crosstalk/genetic interaction of hypoxia/oxygen and JA in the first paragraph.

2) It is interesting that submergence can affect histone modifications on the OPR3 locus shown in Fig 4b,c. However, I think the genetic evidence is needed to show whether the change of histone modification is related to the crosstalk of submergence and wound response. Histone H3K4 methyltransferase SDG2 or ATX1/2 or the JMJC family genes that contribute to H3K4 demethylation could be candidates to be tested. Or alternatively, chemical treatments with histone acetylation or deacetylation inhibitors could be another choice. Such genetic evidences can provide the solid conclusion that histone modifications are involved.

3) Lines 160-163, the authors indicated that SA might mediate the crosstalk between hypoxia/submergence and JA. I think that this is an interesting discussion. I would like to suggest the authors to add more explanation with references.

Reviewer #3 (Remarks to the Author):

Under normal conditions, wounding induces via the JA pathway responses which help to defend herbivore attack. The authors present evidence that submergence of the model plant Arabidopsis results in no/less activation of the JA pathway by wounding. This manuscript also reports that

submergence affects histone modification which might enable the plant to memorize the stress event. The authors discuss that the submergence-induced repression of the wound response contributes to increased plant damage by flooding in agriculture.

This is an original research topic and the data provided are new.

General:

1. Since the authors are emphasizing the relevance in agriculture it would be relevant to know whether similar effects are detectable in economically important plants.
2. The authors could connect their results to a broader context. In rice it is known that the ethylene pathway is activated by flooding and responsible for the majority of submergence-induced responses. In Arabidopsis it is known that ethylene inhibits the MYC2-VSP-branch of JA signalling. Therefore it would be obvious to test whether activation of the ethylene pathway inhibits the JA pathway upon submergence in Arabidopsis.
3. Savchenko et al. 2019 showed that OPDA levels (a precursor of JA) increase after waterlogging and that oxylipins of the AOS and HPL pathway contribute to survival upon waterlogging and submergence (mutants show lower survival). This should be included in the discussion.

Details/minor points:

4. Were all leaves wounded? According to the schematic figures not all leaves were wounded, in suppl. Fig 3A only the first leaves are wounded, the methods describe wounding of cotyledones and first leaves.
 5. Page 3 line 56: JA-Ile binds to COI1 not JA
 6. Page 3 line 57: correct would be: SCFCOI1-JAZ co-receptor complex
 7. Page 4 line 62: rephrase: „Plant leaves contain epidermal cells“
 8. Page 6 line 101: to the JA pathway
 9. Fig. 2: The effect of 2 h argon on wound-induced gene expression seems to be smaller (2e, factor 2) than 1h submergence (e.g. 2b factor >4)
 10. Fig. 2: why is the relative expression in 2e about tenfold higher than in comparable samples in Fig. 2b?
 11. Of AOCs and LOXs there are several isozymes. e.g. according to Chauvin et al 2016 all four LOXs contribute to the wound response. Why were only AOC2 and LOX2 tested and not AOS and the other AOCs or LOXs?
 12. Suppl Fig. 3: please explain DW in the legend
- Several references (12, 13 and 15) are suboptimal, maybe there is some confusion with numbering? e.g. Reference 13 is not appropriate, use e.g. Thines et al. 2007 or Chini et al. 2007
13. Page 4 line 69 change „levels“ to „induction“.

Response to reviewer 1

1. In general, this work focuses only on a very small set of genes that are involved in wounding/ pathogen response. However, there are many more factors, as well as wound-responsive genes at different stages of the response, which are ignored here. I miss an analysis of wound-responsive genes, despite JA metabolism and action, for example PR proteins or PDFs. Also, the wound response has certain timing, and you only look at a 1-h timepoint.

Response: We additionally analyzed wound-responsive genes as suggested (see **Fig. 4, Supplementary Fig. 4**). Also, we performed time-course analysis of wound responses (see **Fig. 1e**).

2. This work could make use of existing transcriptome studies on hypoxia and submergence. It remains unclear why you chose those few genes at all, and not others. How is their expression in other hypoxia/ submergence experiments? Are those the only pathogen-related genes differentially expressed? A short look into published data does not reveal those few genes as mostly affected. Indeed, these genes are often not regulated in similar stress studies.

Response: In this study, we analyzed wound-responsive genes which are related with JA biosynthesis or JA responses. Those genes were not responsive to hypoxia or submergence (**Fig. 1**). We found that the wound-responsive genes become less sensitive to wound after plants have undergone submergence. To the best of our knowledge, it is the first report to analyze that expression of wound-responsive genes is changed after submergence. Because previous reports only treated hypoxia or submergence, expression of the wound-responsive genes including *JAZs* and *OPR3* would be not changed. We chose *JAZs* and *OPR3* genes as representative wound-responsive JA responsive genes and a JA biosynthesis gene, respectively (Acosta *et al.*, 2013, *Proc. Natl. Acad. Sci. U. S. A.* **110**: 15473-15478; Koo *et al.*, 2009, *Plant J.* **59**: 974-986).

3. Another major problem is the timing of stress. Obviously, the authors see an effect already after 1 min of submergence, and used a treatment up to 1 h. However, this is not a natural

situation, where flooding often occurs for many hours or days. You need to include more time-points (e.g., 24 and 48 h) to get a realistic situation.

Response: We totally agree with the comment. We thus treated submergence up to 48 h and analyzed wound responses after submergence (see **Fig. 1f**).

4. The authors claim that low-oxygen/ hypoxia might be involved. However, after 1 min, there is no hypoxia in submerged leaves, even if submergence was done in darkness. Within 1 h of submergence in darkness, oxygen levels decline, but not up to severe hypoxia (see Lee et al. 2011, Vashisht et al. 2011). If submergence is done in illumination, there is not hypoxia at all. The authors do not specify the light levels during submergence and of control plants, and they also do not mention at what time (related to start of daylight) the stress was applied.

In line with this, if you expect hypoxia, you should demonstrate this by analyzing for example *ADH1* expression in your submergence treatments, as you did for the hypoxia gassing.

Response: Thanks for the critical comment. As the comment, up to 1 h submergence did not trigger hypoxia in plants. We confirmed it via analyzing expression of *ADH1* gene, which is a molecular marker for hypoxia (**Fig. 4a**). We thus concluded that hypoxia is not the main reason for reduced wound responses after short-term submergence. Alternatively, we found that early submergence signal, ethylene, is involved in these responses. Submergence-mediated suppression of wound responses were partially restored in the ethylene-insensitive mutant, *ein2-1* (**Fig. 4c**). Based on the new results, we changed descriptions in the manuscript.

5. Another plant hormone that is important under submergence and increases much faster than oxygen declines is ethylene. This would be a much more obvious signal for plants, rather than oxygen deficiency. Interaction between ethylene and JA might be present.

Response: To analyze role of ethylene, we used the ethylene-insensitive *ein2-1* mutant. We found that submergence-mediated suppression of wound responses was restored in the mutant, suggesting that ethylene is involved in these responses (see **Fig. 4**).

6. The experiments are sometimes not easy to understand. The drawings are helpful, but not always accurate, and the Mat&Meth section is also not complete. For example, lines 179-180 only state wounding after de-submergence, but some data apparently are from wounding under water. How was the timing there? How long were the plants under water before wounding in water? Only a few seconds, or a few minutes? In Fig. 1b, do you also have an air/re-air time point (i.e., 2 h after wounding)? The wounding response quickly declines, and maybe it already went down during this time.

Response: Because the duration of submergence was different in some experiment, we described brief experimental procedures in **figure legends**. In most experiments, plants were submerged for 1 h in water. In **Fig. 1b**, plants were wounded in water and incubated for 1 h in water and then 1 h in air. Therefore, we harvested plants 2 h after wounding. The wound responses were declined, thus expression of *JAZ7* was only marginally increased. However, we could observe wound responses by analyzing expression of *JAZ10* and *OPR3*, which was still increased at 2 h after wounding.

7. The manuscript ignores previous publications that also deal with pathogens and submergence, and that both show a positive effect of submergence on pathogen tolerance. Hsu et al. 2013 demonstrate enhanced bacterial resistance after flooding, and Baumler et al. 2019 describe submergence-induced expression of *ERF#111*, a gene that strongly responds to wounding.

Response: We included descriptions on the effects of submergence on pathogen tolerance based on Hsu et al., 2013 (see **line 244 to 247**). In addition, we analyzed expression of *ERF#111/ABRI* gene (see **Fig. 4b**).

8. Regarding ChIP assays, should you not use another negative control, i.e. a promoter that does not respond to your treatment, rather than relate the quantification to the input fraction?

Response: We included a negative control using the *ACT7* locus for ChIP assays (see **Fig. 5**). While H3K4me2 levels at the *ACT7* locus were decreased by submergence in Mock-treated plants, those were not altered in other treatments. These results suggest that H3K4me2 levels are reduced at the *OPR3* locus at least in wounded plants after submergence. On the

other hand, H3Ac levels at the *ACT7* locus were not altered by any treatment, suggesting that H3Ac levels at the *OPR3* locus are reduced by submergence.

9. Line 158, please specify "pathogen infection" (i.e., which pathogens).

Response: We changed the descriptions based on other references (Hsu *et al.*, 2013, *Plant Cell* **25**: 2699-2713). We specified that plants become resistant to bacterial pathogen *Pseudomonas syringae* after submergence (see **line 246**).

10. Materials and Methods, did your MS medium contain sugars? Were the plates placed vertically or horizontally in the growth chambers? How was the positioning during submergence? How much agar was in the medium? Was submergence and hypoxia treatment in light or darkness?

Response: Our MS-agar medium contains 0.7% agar and no sucrose. Plates were placed horizontally in the growth room. During submergence, plates were placed horizontally in the water-filled transparent box without lids. Submergence and hypoxia treatments were performed in the growth room in light. We added detailed descriptions in the **Methods** section.

11. Your transgenic lines (line 16, line 225), which generation did you use? How many independent lines did you use?

Response: We obtained three independent second generation (T2) transgenic plants and confirmed GUS or iGluSnFR expressions in all transgenic lines. We used a single line for detailed analysis to observe effects of submergence. We added detailed descriptions in Method section (see **line 335-338** and **line 371-374**).

12. In Figure S3, you did not do ANOVA although these are similar data as in the main manuscript. Please use appropriate statistics throughout the figures. Figure S3d, what does "DW" mean? Did you use a mock spraying treatment?

Response: We changed statistical analysis using ANOVA test in **Supplementary Fig. 3**. “DW” indicates a mock spraying treatment. We added descriptions on “DW” in the figure legend.

Response to reviewer 2

1. The authors may add more introductions to the current knowledge on the crosstalk/genetic interaction of hypoxia/oxygen and JA in the first paragraph.

Response: We added introductory paragraphs to the previous studies on the interaction between hypoxia/submergence and ethylene. Also, effects of hypoxia/submergence on histone modifications were also added in the **Introduction** section.

2. It is interesting that submergence can affect histone modifications on the *OPR3* locus shown in Fig 4b,c. However, I think the genetic evidence is needed to show whether the change of histone modification is related to the crosstalk of submergence and wound response. Histone H3K4 methyltransferase *SDG2* or *ATX1/2* or the *JMJC* family genes that contribute to H3K4 demethylation could be candidates to be tested. Or alternatively, chemical treatments with histone acetylation or deacetylation inhibitors could be another choice. Such genetic evidences can provide the solid conclusion that histone modifications are involved.

Response: Because we cannot test all histone methylation/acetylation-related mutants, we examined wound responses after submergence in a histone deacetylase (*axe1-5*), a histone demethylase (*jmj30-2*), and histone methyltransferases (*atx1-2 atx2-1*). However, we cannot find any difference in wound responses in these mutants. Alternatively, we treated trichostatin A (TSA), which is a histone deacetylase inhibitor to examine the role of histone deacetylases. While expression of the *OPR3* and *JAZ10* genes in the seedlings that had undergone submergence was not altered by TSA treatment until 1 h after wounding, TSA-treatment slightly rescued wound responses at 2 h after wounding (see **Fig. 5d**). These results suggest that histone acetylation and methylation might redundantly control wound responses

after submergence, thus a single alteration of histone modification cannot fully rescue these responses.

3. Lines 160-163, the authors indicated that SA might mediate the crosstalk between hypoxia/submergence and JA. I think that this is an interesting discussion. I would like to suggest the authors to add more explanation with references.

Response: We added detailed descriptions on JA-SA cross-talks after submergence (see **line 236-250**).

Response to reviewer 3

General

1. Since the authors are emphasizing the relevance in agriculture it would be relevant to know whether similar effects are detectable in economically important plants.

Response: Previous studies showed that flooding triggers increase in herbivore populations in rice. In addition, feeding activities of insect herbivores are increased in citrus. We added the related descriptions in **Introduction** section (see **line 47-50**). To the best of our knowledge, it is first time to analyze effects of submergence on wound-induced JA biosynthesis.

2. The authors could connect their results to a broader context. In rice it is known that the ethylene pathway is activated by flooding and responsible for the majority of submergence-induced responses. In Arabidopsis it is known that ethylene inhibits the MYC2-VSP-branch of JA signalling. Therefore it would be obvious to test whether activation of the ethylene pathway inhibits the JA pathway upon submergence in Arabidopsis.

Response: We additionally analyzed effects of submergence on JA and ethylene responses and found that expression of ethylene-responsive genes was rapidly increased by

submergence. Next, we used the ethylene-insensitive *ein2-1* mutant and found that submergence-mediated suppression of wound responses were partially rescued in the *ein2-1* mutant, suggesting that ethylene is involved in these responses (see **Fig. 4**).

3. Savchenko et al. 2019 showed that OPDA levels (a precursor of JA) increase after waterlogging and that oxylipins of the AOS and HPL pathway contribute to survival upon waterlogging and submergence (mutants show lower survival). This should be included in the discussion.

Response: We added descriptions based on Savchenko et al., 2019 in **Discussion** section (see **line 251-258**).

Details/minor points:

4. Were all leaves wounded? According to the schematic figures not all leaves were wounded, in suppl. Fig 3A only the first leaves are wounded, the methods describe wounding of cotyledones and first leaves.

Response: For gene expression analyses and ChIP assays, both cotyledons and first leaves were wound. For histochemical GUS and DAB staining, only first leaves were wounded to show local wound responses. We added detailed procedures in **Method** section.

5. Page 3 line 56: JA-Ile binds to COI1 not JA.

Response: We corrected it as suggested (see **line 59**).

6. Page 3 line 57: correct would be: SCFCOI1–JAZ co-receptor complex

Response: We corrected it as suggested (see **line 61**).

7. Page 4 line 62: rephrase: „Plant leaves contain epidermal cells“

Response: We corrected it (see **line 101-102**).

8. Page 6 line 101: to the JA pathway

Response: We corrected it as suggested (see **line 127**).

9. Fig. 2: The effect of 2 h argon on wound-induced gene expression seems to be smaller (2e, factor 2) than 1h submergence (e.g. 2b factor >4)

Response: We agree with the comment. We thus suspected that hypoxia would be not the main reason for the submergence-mediated suppression of wound responses. Additional analyses found that ethylene is involved in these responses (see **Fig. 4**). We added figures and re-described the text.

10. Fig. 2: why is the relative expression in 2e about tenfold higher than in comparable samples in Fig. 2b?

Response: In previous Figure 2b, expression of *JAZ10* and *OPR3* genes at “0 h reaeration” was set to 1. The “0 h reaeration” indicates that plants were submerged for 1 h and then wounded right after the reaeration. On the other hand, expression of genes in “Mock-Air” was set to 1 in previous Figure 2e. The “Mock-Air” indicates that plants were not wounded and submerged.

By doing review processes, we decided to remove previous Figure 2b because we found some technical errors in this assay. We re-described text according to the newly organized figures.

11. Of AOCs and LOXs there are several isozymes. e.g. according to Chauvin et al 2016 all four LOXs contribute to the wound response. Why were only AOC2 and LOX2 tested and not AOS and the other AOCs or LOXs?

Response: We analyzed more *AOCs* and *LOXs* as suggested (see **Supplementary Fig. 2**).

12. Suppl Fig. 3: please explain DW in the legend

Response: We added descriptions on “DW” in the figure legends.

13. Several references (12, 13 and 15) are suboptimal, maybe there is some confusion with numbering? e.g. Reference 13 is not appropriate, use e.g. Thines et al. 2007 or Chini et al. 2007

Response: We changed all three references to more appropriate articles (previous 12, 13, and 15 are now 12, 13, 14, respectively).

14. Page 4 line 69 change „levels“ to „induction“.

Response: We changed it as suggested (see **line 108**).

Reviewers' comments:

Reviewer #1 (Remarks to the Author):

In their revision, the authors have partially improved the manuscript. However, the experimental set-up, especially the timing is still unclear to me. Also, some conclusions are not supported by the data. My detailed comments are below.

Line 69, plants actually cannot "choose" the escape or quiescence strategy, at least not most of the species. The genotype gene composition determines which response is chosen under water, as it was shown for rice genotypes.

Line 91, instead of "JA biosynthesis locus", use "OPR3 locus" since many gene products from different loci are involved in JA biosynthesis.

Line 106/107, in your rebuttal letter you used this phrase: "We chose JAZs and OPR3 genes as representative wound-responsive JA responsive genes and a JA biosynthesis gene, respectively (Acosta et al., 2013, Proc. Natl. Acad. Sci. U. S. A. 110: 15473-15478; Koo et al., 2009, Plant J. 59: 974-986)." – why did you not add a phrase like this into the manuscript? This would help to justify the choice of genes analyzed.

I still have a big problem with the timing of experiments in Figure 1. Your additional remarks still are not sufficient and drawings are not correct. Please write more precisely, for example as suggested here (square brackets contain just alternatives or comments since the experiment is really hard to understand):

-For Fig. 1a, line 588f, "... Cotyledons and leaves of the Col-0 seedlings were left in the air (air) or transferred into the water (sub), wounded xxx min later [or immediately?] and then harvested after 1 h incubation. Three biological replicates were averaged."

-For Fig. 1b, line 590f, "... The Col-0 seedlings were wounded in the water immediately [correct? Or after xx min?] after submergence and incubated for 1 h. Submerged seedlings were then re-aerated for 1 h before they were harvested (Sub-air) or were left in water for another hour before harvest (Sub). Three biological replicates were averaged."

-For Fig. 1c, line 594f, "... Re-aerated seedlings were immediately [correct?] wounded and then harvested after 1 h."

-If I understood those experiments correctly, then the drawing for Fig. 1b is wrong, since you had three variants, one without submergence that is not in the sketch, and one without re-aeration that is not in the sketch.

Line 190, this experiment is good, but you measured ADH1 expression only after 1 h of re-aeration, not directly after submergence, as far as I understood. I agree that you should have no hypoxia even directly under submergence, but you do not show the correct data for this statement. Re-aeration leads to a quick down-regulation of hypoxia marker genes, so one should measure ADH1 expression directly after submergence.

Line 201, this statement is vague and does not fit well with your data in Fig. 4c. Only JAZ7 seems to respond differently in *ein2-1*, but all other genes are not different compared to wildtype. You need to clearly describe those results in the text, and also in the discussion (line 264).

Experiments with histone modifications, I did not understand how you normalized the data to "% of input". What measure did you use as the "input"? The methylation at the control Actin locus also responds to submergence. In Fig. S5, you do not show the Actin data. Those are required as a control.

You still used several t-tests, where ANOVA would have been more appropriate, i.e., Fig. 3b, Fig. 4a, Fig. 5d.

You provide raw data for main figures, which is good, but you forgot raw data for the Supplemental Figures.

Regarding the feeding experiment: as I understood, you put 20 caterpillars on 10 plants (2 on each plant), but you only collected 7 and 8 (not 10, as the legend states, according to raw data). What happened to the other caterpillars? Did they die or run away? Could you repeat this experiment with *ein2-1*? This would show whether ethylene is important for the defense response.

Lines 271ff, this part is pure speculation. You do not have any data on this. You should avoid phrases like "might be the reason", rather use "could be the reason" or "we hypothesize that...".

Fig. 1, expression levels in 1a vs. 1c, can you explain the strong difference in expression values?

Fig. 1f, should it be "time of submergence" instead of "time after submergence"?

The text style still requires improvements, as listed here (I probably did not find all problems).

-Line 31, instead of "even under normoxia conditions" use this phrase "without signs for lack of oxygen". Next sentence should be also re-phrased.

-Line 65, instead of "for submergence" use another phrase, for example "in response to submergence" or "under".

-Line 278/279, rephrase sentence. A word is missing and normoxia is mis-spelled.

-Line 289, avoid a phrase like this "we cannot find the key genes", rather write in impersonal style.

-Line 361, "spray-treated" is not a commonly used word.

-Line 410/411, please re-phrase.

Reviewer #2 (Remarks to the Author):

The authors have addressed most of my concerns. I have no further suggestions.

Reviewer #3 (Remarks to the Author):

The manuscript has strongly improved by Revision. I only have one comment:

Page 5 line 91: „acetylation at the JA biosynthesis gene locus“ add OPR3 because there are several loci for JA biosynthetic enzymes.

Response to reviewer #1

1. Line 69, plants actually cannot "choose" the escape or quiescence strategy, at least not most of the species. The genotype gene composition determines which response is chosen under water, as it was shown for rice genotypes.

Response: we revised the sentence (see line 69).

2. Line 91, instead of "JA biosynthesis locus", use "OPR3 locus" since many gene products from different loci are involved in JA biosynthesis.

Response: we revised the sentence as suggested (see line 91).

3. Line 106/107, in your rebuttal letter you used this phrase: "We chose JAZs and OPR3 genes as representative wound-responsive JA responsive genes and a JA biosynthesis gene, respectively (Acosta et al., 2013, Proc. Natl. Acad. Sci. U. S. A. 110: 15473-15478; Koo et al., 2009, Plant J. 59: 974-986)." – why did you not add a phrase like this into the manuscript? This would help to justify the choice of genes analyzed.

Response: we added the sentence as suggested (see line 106-108).

4. I still have a big problem with the timing of experiments in Figure 1. Your additional remarks still are not sufficient and drawings are not correct. Please write more precisely, for example as suggested here (square brackets contain just alternatives or comments since the experiment is really hard to understand):

4-1. For Fig. 1a, line 588f, "... Cotyledons and leaves of the Col-0 seedlings were left in the air (air) or transferred into the water (sub), wounded xxx min later [or immediately?] and then harvested after 1 h incubation. Three biological replicates were averaged."

Response: we described more precisely as suggested (see line 613-615).

4-2. For Fig. 1b, line 590f, "... The Col-0 seedlings were wounded in the water immediately [correct? Or after xx min?] after submergence and incubated for 1 h. Submerged seedlings were then re-aerated for 1 h before they were harvested (Sub-air) or were left in water for another hour before harvest (Sub). Three biological replicates were averaged."

Response: we revised the sentence to describe more detailed method (see line 616-619).

4-3. For Fig. 1c, line 594f, "... Re-aerated seedlings were immediately [correct?] wounded and then harvested after 1 h."

Response: we revised the sentence (see line 621-622).

4-4. If I understood those experiments correctly, then the drawing for Fig. 1b is wrong, since you had three variants, one without submergence that is not in the sketch, and one without re-aeration that is not in the sketch.

Response: we corrected the drawings in Fig. 1a, 1b, and 1c.

5. Line 190, this experiment is good, but you measured ADH1 expression only after 1 h of re-aeration, not directly after submergence, as far as I understood. I agree that you should have no hypoxia even directly under submergence, but you do not show the correct data for this statement. Re-aeration leads to a quick down-regulation of hypoxia marker genes, so one should measure ADH1 expression directly after submergence.

Response: we newly performed experiments to analyze *ADH1* expression right after submergence. As anticipated, *ADH1* expression did not exhibit any change after 1 h submergence. We changed the previous Fig. 4a with the new one.

6. Line 201, this statement is vague and does not fit well with your data in Fig. 4c. Only *JAZ7* seems to respond differently in *ein2-1*, but all other genes are not different compared to wildtype. You need to clearly describe those results in the text, and also in the discussion (line 264).

Response: in our statistical analysis, expression of *AOC2*, *JAZ7*, and *VSP1* in submergence-treated *ein2-1* mutant was significantly higher than that in submergence-treated Col-0 seedlings (Fig. 4c). These data indicate that EIN2 affects expression of several genes under submergence conditions, thus we described that gene expressions were partially restored by *ein2-1* mutation. We revised the text to clearly describe the experimental results (see line 203-207 and line 273-274).

7. Experiments with histone modifications, I did not understand how you normalized the data to "% of input". What measure did you use as the "input"? The methylation at the control Actin locus also responds to submergence. In Fig. S5, you do not show the Actin data. Those are required as a control.

Response: we added more detailed descriptions on the ChIP assay method and input (see line 404-415). Because H3K4me2 was also altered by submergence at the control *ACT7* locus, it seems that changes of H3K4me2 in mock-treated seedlings at *OPR3* locus would be a technical error. We described this issue in the text (see line 213-221). Finally, we added Actin data in Supplementary Fig. 5.

8. You still used several t-tests, where ANOVA would have been more appropriate, i.e., Fig.

3b, Fig. 4a, Fig. 5d.

Response: we adopted ANOVA test in all Figures.

9. You provide raw data for main figures, which is good, but you forgot raw data for the Supplemental Figures.

Response: we uploaded raw data for the Supplemental Figures.

10. Regarding the feeding experiment: as I understood, you put 20 caterpillars on 10 plants (2 on each plant), but you only collected 7 and 8 (not 10, as the legend states, according to raw data). What happened to the other caterpillars? Did they die or run away? Could you repeat this experiment with *ein2-1*? This would show whether ethylene is important for the defense response.

Response: we put 20 caterpillars on 10 plants, but only found 8~9 caterpillars on plants after 5 days, possibly because several of them moved away from the plants. We added descriptions on this issue in the text and corrected the figure legend (see line 146-147 and line 651).

Submergence-mediated suppression of wound responses including expression of *AOC2*, *JAZ7*, and *VSP1* genes was partially restored by *ein2-1* mutation, but still expression of most JA biosynthesis and responsive genes was inhibited by submergence in *ein2-1* mutant. These results indicate that mutation of *EIN2* is not sufficient to fully restore wound responses. Synergistic action of ethylene and other unknown regulators would possibly be required to inhibit wound responses upon submergence. We described this idea in the **Discussion** section (line 287-299). Because caterpillar feeding assays are highly variable, further works will be required to find major factors and obtain clear results. Therefore, we decided not to perform herbivore feeding assays with *ein2-1* mutant. We hope the reviewer would accept our reasoning on this affair.

11. Lines 271ff, this part is pure speculation. You do not have any data on this. You should avoid phrases like "might be the reason", rather use "could be the reason" or "we hypothesize that...".

Response: we changed the phrase into "could be the reason" (see line 285).

12. Fig. 1, expression levels in 1a vs. 1c, can you explain the strong difference in expression values?

Response: The extent of wound-induced expression of JA-responsive genes was different in these assays, possibly because plant wound responses depend on timing of treatment during the day (Goodspeed et al., *Arabidopsis* synchronizes jasmonate-mediated defense with insect circadian behavior, *PNAS*, 109: 4674-4677, 2012). While each set of

experiments was performed at different time of the day, we treated “Air-mock”, “Air-wound”, “Sub-mock”, and “Sub-wound” simultaneously in each experiment. Therefore, the expression patterns were similar in these assays.

13. Fig. 1f, should it be "time of submergence" instead of "time after submergence"?

Response: we corrected the phrase as commented.

14. The text style still requires improvements, as listed here (I probably did not find all problems).

14-1. Line 31, instead of "even under normoxia conditions" use this phrase "without signs for lack of oxygen". Next sentence should be also re-phrased.

Response: we changed the phrase as suggested. Next sentence was also re-phrased (see line 30-35).

14-2. Line 65, instead of "for submergence" use another phrase, for example "in response to submergence" or "under".

Response: we changed the phrase as suggested (see line 65).

14-3. Line 278/279, rephrase sentence. A word is missing and normoxia is mis-spelled.

Response: we rephrased the sentence (see line 287-288).

14-4. Line 289, avoid a phrase like this "we cannot find the key genes", rather write in impersonal style.

Response: we rephrased the sentence (see line 301-302).

14-5. Line 361, "spray-treated" is not a commonly used word.

Response: we used “ ... were sprayed with ...” in the text (see line 371-374).

14-6. Line 410/411, please re-phrase.

Response: we re-phrased the sentence (see line 431-433).

Response to reviewer #3

1. The manuscript has strongly improved by Revision. I only have one comment:

Page 5 line 91: „acetylation at the JA biosynthesis gene locus“ add OPR3 because there are several loci for JA biosynthetic enzymes.

Response: we added “OPR3” as suggested (**see line 91**).

REVIEWERS' COMMENTS:

Reviewer #1 (Remarks to the Author):

After the second revision, the manuscript improved a lot in terms of easier reading and understanding. I only have some minor comments.

Line 85, remove "the" (6th word).

Line 235, the difference after 2 h is not as dramatic as the text sounds. Maybe re-phrase like this: "However, wounding slightly but significantly elevated expression ..., while changes upon wounding were not significant in mock-treated seedlings (Fig. 5d)."

Line 239, first word in line should start with small letter.

Line 286, "at the molecular level"

Line 301, "raising the possibility that an EIN2-independent"

Fig. 4a, indicate submergence duration for ADH1 expression panel in y-axis.

Response to reviewer #1

After the second revision, the manuscript improved a lot in terms of easier reading and understanding. I only have some minor comments.

1. Line 85, remove "the" (6th word).

Response: We added “the” in the text (**see line 84**).

2. Line 235, the difference after 2 h is not as dramatic as the text sounds. Maybe re-phrase like this: "However, wounding slightly but significantly elevated expression ..., while changes upon wounding were not significant in mock-treated seedlings (Fig. 5d)."

Response: We revised the sentence as suggested (**see line 230 to 233**).

3. Line 239, first word in line should start with small letter.

Response: We corrected the error (**see line 234**).

4. Line 286, "at the molecular level"

Response: We corrected the phrase as suggested (**see line 282**).

5. Line 301, "raising the possibility that an EIN2-independent"

Response: We corrected the phrase as suggested (**see line 296**).

6. Fig. 4a, indicate submergence duration for ADH1 expression panel in y-axis.

Response: We indicated submergence duration in **Figure 4a**.